# Impact of Nucleic Acid Amplification Test on Clinical Outcomes in Patients with *Clostridioides difficile* Infection

**DOI:** 10.3390/antibiotics12030428

**Published:** 2023-02-21

**Authors:** Yota Yamada, Motoyasu Miyazaki, Hisako Kushima, Yukie Komiya, Akio Nakashima, Hiroshi Ishii, Osamu Imakyure

**Affiliations:** 1Department of Pharmacy, Fukuoka University Chikushi Hospital, Fukuoka 818-8502, Japan; 2Department of Infection Control and Prevention, Fukuoka University Chikushi Hospital, Fukuoka 818-8502, Japan; 3Department of Respiratory Medicine, Fukuoka University Chikushi Hospital, Fukuoka 818-8502, Japan; 4Department of Clinical Laboratory, Fukuoka University Chikushi Hospital, Fukuoka 818-8502, Japan

**Keywords:** *Clostridioides difficile*, nucleic acid amplification test, clinical outcome

## Abstract

A nucleic acid amplification test (NAAT) is recommended to determine whether or not patients have a *Clostridioides difficile* infection (CDI) when the glutamate dehydrogenase activity assay is positive and the rapid membrane enzyme immunoassays for toxins is negative. In our hospital, a NAAT was introduced to diagnose CDI precisely in April 2020. This study aimed to investigate the impact of a NAAT on the clinical outcomes in patients with CDI at our hospital. Seventy-one patients diagnosed with CDI between April 2017 and March 2022 were included in our study. Patients with CDI were divided into two groups: before (pre-NAAT) and after (post-NAAT) the introduction of NAAT. The clinical outcome was compared between the two groups. Of the 71 patients with CDI, 41 were sorted into the pre-NAAT group and 30 into the post-NAAT group. The clinical cure rate was significantly higher in the post-NAAT group compared to the pre-NAAT group (76.7% vs. 48.8%, *p* = 0.018). In the multivariable analysis, the clinical cure was significantly associated with the introduction of NAAT (*p* = 0.022). Our findings suggest that the introduction of NAAT can improve the clinical outcomes in CDI patients.

## 1. Introduction

One of the most common anaerobic bacteria causing healthcare-associated diseases is *Clostridioides difficile* (CD), which is also known to cause several infections, including infectious diarrhea and pseudomembranous enterocolitis [1]. Over the previous decade, the incidence and severity of *Clostridioides difficile* infection (CDI) has increased considerably, which is concomitant with the prevalence of hypervirulent bacterial strains [2,3,4]. All-cause 30-day mortality rates with CDI patients have been reported to range from 11.8% to 23.0% [5]. Older age, low serum albumin (Alb) levels, and high white blood cell (WBC) counts were associated with increased mortality in patients with CDI [6,7]. Even with appropriate treatment for patients with CDI, about 30% of the patients relapse [8,9,10]. Older age, renal failure, and the use of a proton pump inhibitor (PPI) are associated with increased risk factors for the recurrence of CDI [11,12]. A recurrence of CDI is estimated to result in medical costs of over JPY 1.28 million (USD 12,600) and an extended hospital stay of 3 weeks [13].

Toxigenic culture (TC), cell cytotoxicity neutralization assay (CCNA), glutamate dehydrogenase (GDH), toxin A/B (TOXIN) enzyme immunoassay (EIA), and nucleic acid amplification testing (NAAT) are among the laboratory tests used to diagnose CDI. These tests have varying degrees of sensitivity and specificity [14,15,16,17]. Conventionally, a two-step algorithm has been proposed, in which the TOXIN production of the isolate is confirmed by CCNA or TC in the case of GDH(+)/TOXIN(−) on a rapid membrane EIA [18]. However, the CCNA and TC require several days to reach a conclusion and are often not carried out in general medical institutions. EIA has the advantage of high specificity, but its sensitivity is insufficient, making the diagnosis of CDI on its own difficult. Although NAAT is more expensive than the other methods, it has been adopted because of its high sensitivity, and it provides a more reliable diagnosis than traditional, less sensitive EIA [16]. NAAT evaluates toxin production, thus avoiding unnecessary treatment for CD that does not produce toxins. In addition, a rapid diagnosis with NAAT allows treatment to be initiated earlier than that with CCNA or TC [16,17]. Undertaking a NAAT immediately following EIA provides the most useful diagnostic information [17] and is the recommended diagnostic algorithm in Japan [19]. However, NAAT is still not widely used in general hospitals in Japan because its introduction has been limited to facilities with additional measures for infection prevention. Therefore, only some reports have examined the impact of the NAAT introduction on clinical outcomes in patients with CDI [20].

Fukuoka University Chikushi Hospital introduced NAAT in 2020 to help diagnose CDI. It is unclear whether the introduction of NAAT has actually increased the diagnostic rate of CDI or improved outcomes for CDI patients. The objective of our study was to compare the clinical outcome before and after the introduction of NAAT in patients with CDI at our hospital.

## 2. Materials and Methods

### 2.1. Study Population

This retrospective cohort study was conducted at the Fukuoka University Chikushi Hospital in Fukuoka, Japan, between April 2017 and March 2022. The Ethics Committee of the Fukuoka University School of Medicine approved the study protocol (No. C22-09-002). Patients who were diagnosed with CDI and started on treatment with anti-CD agents were included in the study. The study excluded participants under the age of 18, who underwent outpatient EIA without being hospitalized, who were not examined via a stool sample, or who were transferred to another hospital during treatment. For patients who had multiple episodes of CDI during the study period, only first episode was included in the analysis. Patients who were transferred to another hospital during treatment and who were not treated with anti-CD agents were excluded from the evaluation of outcomes but included when evaluating the diagnostic rate.

### 2.2. Tests and Diagnosis of CDI

The CDI testing algorithm consists of an initial screening step using a premier EIA for GDH, followed by a NAAT (Cepheid, Xpert^®^
*C. difficile*, US) for GDH(+)/TOXIN(−). *C. DIFF* QUIK CHEK COMPLETE^®^ (TechLab, Blacksburg, VA, USA), a rapid membrane EIA was used to detect both the pathogen’s GDH and TOXIN. The Cepheid NAAT is a real-time polymerase chain reaction assay targeting the toxin B genes. If the CDI test alone does not confirm the diagnosis of CDI, it is recommended that the diagnosis of CDI should be made by clinical symptoms [19]. Even if a physician tests a patient for CDI, if the diarrhea symptoms are temporary or if there is no fever or abdominal pain, the physician will explore the possibility of a disease other than CDI. A patient with GDH(+) TOXIN(+) by EIA was defined as non-CDI if the attending physician determined that the patient was a carrier of CD or if they suspected another disease.

### 2.3. Clinical Characteristics

Patient data were collected from electronic medical records and studied retrospectively. Clinical characteristics included age, sex, use of (PPI) and probiotics, inflammatory bowel disease (IBD) as underlying diseases, chemotherapy for neoplasm, the severity of the disease, treatment with anti-CD agents, length of hospital stay, length of treatment, and clinical outcome. The Charlson comorbidity index (CCI) was used to measure comorbidity [21], and the severity of CDI was assessed using the MN criteria, which was originally proposed in Japan [22]. In order to assess the CDI severity, we investigated age, abdominal symptoms, body temperature, times of diarrhea, WBC count, estimated glomerular filtration rate, serum Alb level, and the presence of imaging findings. The risk of each item for CDI severity was rated on a scale of 0–3 points, and severity was rated by summing each score (Appendix A). According to the total score, severity was classified as mild (0–4), moderate (5–9), severe (10–13), and critical (14–19). Based on previous reports, we divided the results into two groups: non-severe (mild to moderate) group and severe (severe to critical) group [22]. A supplemental table (Appendix A) shows the actual severity judgments made for the 71 patients in this study.

### 2.4. Clinical Outcome

Clinical outcome was assessed by clinical cure rate, recurrence rate, and 30-day mortality. Clinical cure was defined as completion of treatment with anti-CD agents within 14 days and absence of diarrhea for at least two consecutive days after completion of treatment. Recurrence was defined as the onset of CDI within 8 weeks of the previous episode. Thirty-day mortality was defined as mortality within thirty days after the diagnosis of CDI.

### 2.5. Statistical Analysis

Clinical characteristics were expressed as numbers with proportions for categorical variables and as median and interquartile range (IQR) for continuous variables. Patients with CDI were divided into two groups: before (pre-NAAT) and after (post-NAAT) the introduction of NAAT. For univariable analysis, the Chi-square or Fisher’s exact test (as appropriate) was applied for proportions and the Mann–Whitney U test was used for medians. Multivariable logistic regression analysis was performed on factors associated with clinical cure in univariable analysis (*p* < 0.2). All statistical analyses were performed using JMP^®^ 10 (SAS Institute Inc., Cary, NC, USA). A *p* value < 0.05 was considered to be significant.

## 3. Results

### 3.1. Patient Characteristics

Of the 197 patients with GDH(+), 71 patients diagnosed with CDI were enrolled in this study (Figure 1). The median age of these patients was 80 (IQR: 68–87) years, and 37 (52.1%) of them were male. Of these 71 patients, 12 (16.9%) had IBD, 47 (66.2%) had received PPI, 48 (67.6%) had received probiotics, and 7 (9.9%) had received chemotherapy for a neoplasm. The median CCI was 2 (IQR: 1-3), with a maximum score of 10. The median (IQR) laboratory values were as follows: 9300/μL (6400–16,100) for WBC counts, 2.4 g/dL (2.0–2.9) for serum Alb levels, and 72.2 mL/min/1.73 m^2^ (45.3–107.5) for estimated glomerular filtration rate. According to the severity classification defined by the MN criteria, 15 (21.1%) patients had a mild disease, 44 (61.9%) patients had a moderate disease, 10 (14.1%) patients had a severe disease, and 2 (2.8%) patients had a critical disease. All patients were treated with anti-CD agents: 56 with metronidazole (MNZ), 13 with vancomycin (VCM), and 2 with both.

There were no significant differences in the clinical characteristics between the pre-NAAT and the post-NAAT group (Table 1).

### 3.2. Diagnostic Rate of CDI in Patients with GDH(+)

As shown in Figure 1, in patients with TOXIN (+) in EIA, the diagnosis rate was 92.7% (38/41) in the pre-NAAT group and 100% (15/15) in the post-NAAT group. Meanwhile, in patients with TOXIN (−) in EIA, the post-NAAT group showed a higher diagnostic rate than the pre-NAAT group (45.6% (26/57) vs. 25.0% (21/84)). In addition, the patients who were transferred to another hospital during treatment (11 patients in the pre-NAAT and 5 in the post-NAAT group) were excluded from the study.

### 3.3. Comparison of Clinical Outcome between Pre-NAAT Group and Post-NAAT Group

There were no significant differences in the clinical outcomes, including the length of hospital stay, length of treatment with anti-CD agents, recurrence of CDI, and 30-day mortality, between the pre-NAAT group and the post-NAAT group (Table 2). However, the clinical cure rate was higher in the post-NAAT group compared to the pre-NAAT group (76.7% vs. 48.8%, *p* = 0.018).

Table 3 lists the factors associated with clinical cures in CDI patients. In a multivariable analysis, we determined that the clinical cure was significantly associated with the NAAT introduction (odds ratio (OR) = 3.31; 95% confidence interval (CI) = 1.18–10.03; *p* = 0.022).

## 4. Discussion

The objective of this study was to compare the clinical outcome before and after the introduction of NAAT in patients with CDI at our hospital. The post-NAAT group demonstrated a higher diagnostic rate than the pre-NAAT group in patients with TOXIN (−). Previous reports have shown that the introduction of NAAT improves the diagnostic rate of CDI [23,24,25], and similar results were obtained in our present study.

The clinical cure rate was significantly higher in the post-NAAT group compared to the pre-NAAT group. This indicates that the introduction of NAAT has improved the diagnostic rate of CDI, resulting in successful treatment with the appropriate therapy for CDI. Of the 43 patients who were clinically cured, 11 were classified as mild by MN criteria (3 patients in the pre-NAAT and 8 in the post-NAAT group). Therefore, it is possible that patients with mild CDI, who were considered to have CDI before the introduction of NAAT, were correctly diagnosed and treated after the introduction of NAAT, leading to an improved clinical cure rate. On the other hand, NAAT potentially has the negative aspect of over-diagnosis [26,27], suggesting that over-diagnosis may impact the clinical outcome. However, since 8 (23.5%) of the 34 patients who were positive for toxin B on NAAT were not eventually diagnosed with clinical CDI in the present study, they were unlikely to be over-diagnosed.

The incidence of severe CDI tended to be lower in the post-NAAT group compared to the pre-NAAT group (Table 1). High TOXIN levels of CD can affect the severity of disease [28]. It has been reported that patients with negative EIA but positive NAAT have a less severe disease than those with positive EIA [26,29]. Some reports suggest that severity of CDI by the MN criteria is associated with mortality [22,30]. However, the severity of CDI in the present study did not differ between the clinical cured and non-cured groups (Table 3). This discrepancy may be due to the small number of cases.

Patients taking probiotics had more clinically curative outcomes more often than those not taking probiotics (Table 3). Probiotics have been reported to be effective in preventing the development of CDI [31,32,33]. The guidelines state insufficient evidence for probiotic preparations as a treatment for CDI [19]. There are also reports that probiotics improve diarrhea symptoms caused by CDI [34,35]. Therefore, further research is needed.

In the present study, there were no significant differences in prognostic factors, such as recurrence of CDI and 30-day mortality, before and after the introduction of NAAT. The 30-day mortality rate has been reported to be lower in patients diagnosed with CDI by NAAT than in those diagnosed with CDI by EIA [36]. Guh et al. reported that patients diagnosed by EIA had a higher recurrence rate than those by NAAT, but there was no difference in CDI-related complications or mortality between the two groups [37]. On the other hand, Longtin et al. reported that patients diagnosed by EIA had more CDI-related complications than those by NAAT [38]. Furthermore, it has been reported that there are differences in the clinical characteristics of patients diagnosed by NAAT and those by EIA [39]. Therefore, this suggests that the introduction of NAAT alone is insufficient to improve CDI patients’ prognosis, and further studies are needed on the clinical outcomes in patients with CDI diagnosed by NAAT. As a matter of course, patients with CDI need to be treated with appropriate medications according to the severity of the disease. In our study, all patients were treated with anti-CD agents: 56 with MNZ, 13 with VCM. and 2 with both. Interestingly, of the 12 severe patients, 9 were treated with MNZ, 2 with VCM, and 1 with both. Previous research has shown that the oral administration of VCM is more effective in treating severe CDI cases than MNZ [40,41]. There was a trend toward being treated by VCM in the clinically cured group and by MNZ in the clinical non-cured group, although there were no statistically significant differences (Table 3). Resistance to CD has also been documented, and the clinical cure rates for CDI with MNZ are reported to be 13–20% lower than those for CDI with VCM [40,41]. Therefore, subjects in the present study might be yet to receive the appropriate treatment according to the severity of CDI, i.e., treatment with MNZ for non-severe cases and with VCM for severe cases.

There are several limitations to this study. First, this is a single-center retrospective study with few cases. Second, we should have investigated whether antibiotics have been previously administered to these CDI patients. However, Khanafer et al. noted that antibiotic treatment is not a causative variable for predicting the disease severity, but rather an adjustment factor [42]. In other words, a history of antibiotic use is a risk factor for developing CDI, not a prognostic factor. Thirdly, our hospital has an IBD center, where many IBD patients with a low average age and a low mortality rate are repeatedly admitted and discharged [43]. Fourthly, although TC was originally recommended for CDI diagnosis in patients with GDH(+)/TOXIN(−) in EIA, it may not represent the actual number of CDI patients because some patients were diagnosed without TC before the introduction of NAAT [18]. Finally, some patients were transferred to another hospital during their treatment, because their clinical symptoms became less severe. Therefore, the exclusion of these patients might have affected the clinical cure rates in this study.

Despite these limitations, we believe that this study is significant because only a few reports have examined the impact of NAAT introduction on the clinical outcomes in patients with CDI. Our study suggests that the introduction of NAAT has reduced the number of patients with CDI who are not correctly diagnosed, despite having CDI, and has contributed to improving the clinical cure rate of patients with CDI. We hope that this study will lead to more facilities introducing NAAT and improving the quality of treatment for patients with CDI.

## 5. Conclusions

In this study, the diagnosis rate of CDI increased after the introduction of NAAT as in the previous reports. There were no significant differences in prognostic factors, such as CDI recurrence or 30-day mortality, between the pre- and the post-NAAT groups. However, the clinical cure rate was significantly higher in the post-NAAT group than in the pre-NAAT group. Although this is a single-center, retrospective study with a small number of cases and needs further validation, the results suggest that the introduction of NAAT may lead to improved clinical outcomes for patients with CDI. We hope that this study will lead to an increase in the number of facilities introducing NAAT in the future and will improve the quality of care for patients with CDI.

## Figures and Tables

**Figure 1 antibiotics-12-00428-f001:**
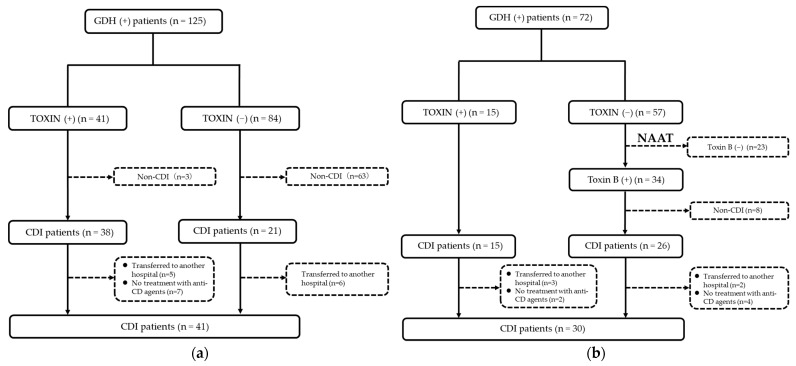
Flow diagram showing the steps of case selection for *Clostridioides difficile* infection (CDI) before the introduction of nucleic acid amplification test (NAAT) (**a**) and after the introduction of NAAT (**b**) (GDH, pathogen’s glutamate dehydrogenase).

**Table 1 antibiotics-12-00428-t001:** Clinical background between pre-NAAT group and post-NAAT group.

Variables	Pre-NAAT (n = 41)	Post-NAAT (n = 30)	*p* Value
Age, median (IQR)	81 (71–87.5)	79.5 (56.5–86.25)	0.59
Male gender, n (%)	20 (48.8)	17 (56.7)	0.51
Severe, n (%)	9 (22.0)	3 (10.0)	0.22
Proton pump inhibitor, n (%)	25 (61.0)	22 (73.3)	0.28
Charlson comorbidity index, median (IQR)	2 (1–3.5)	2 (0–3)	0.48
Inflammatory bowel diseases, n (%)	6 (14.6)	6 (20.0)	0.55
Chemotherapy for neoplasm, n (%)	2 (4.9)	5 (16.7)	0.12
White blood cell count, median (IQR)	9800 (7600–17,450)	9150 (5725–13,500)	0.24
Estimated glomerular filtration rate, median (IQR)	80.3 (44.3–108.7)	70.6 (44.3–106.5)	0.78
Serum albumin level, median (IQR)	2.4 (2.0–2.8)	2.6 (2.1–3.1)	0.21

NAAT, nucleic acid amplification test; CDI, *Clostridioides difficile* infection; and severe, severe to critical.

**Table 2 antibiotics-12-00428-t002:** Comparison of clinical outcome between pre-NAAT group and post-NAAT group.

Variables	Pre-NAAT (n = 41)	Post-NAAT (n = 30)	*p* Value
Number of days in hospital, median (IQR)	33 (25.5–50)	40 (25.5–58.25)	0.56
Number of days of treatment with anti-CD agents, median (IQR)	11 (7.5–14)	10.0 (7.75–14.0)	0.92
Clinical cure, n (%)	20 (48.8)	23 (76.7)	0.018
Recurrence of CDI, n (%)	4 (9.8)	3 (10.0)	1.00
30-day mortality, n (%)	9 (22.0)	3 (10.0)	0.22

NAAT, nucleic acid amplification test; CDI, *Clostridioides difficile* infection.

**Table 3 antibiotics-12-00428-t003:** Factors associated with clinical cure (univariable and multivariable analyses).

Variables		Clinical Non-Cure (n = 28)	Clinical Cure (n = 43)	*p* Value	Multivariable Analysis		*p* Value
					OR	95% CI	
Age, median (IQR)		76.5 (61–88.5)	81.0 (73–87)	0.59	-	-	-
Male gender, n (%)		15 (53.6)	22 (51.2)	0.84	-	-	-
Severe CDI, n (%)		5 (17.9)	7 (16.3)	0.86	-	-	-
NAAT introduction, n (%)		7 (25.0)	23 (53.5)	0.018	3.31	1.18–10.03	0.022
Proton pump inhibitor, n (%)		17 (60.7)	30 (69.8)	0.43	-	-	-
Probiotics, n (%)		16 (57.1)	32 (74.4)	0.13	-	-	-
The number of days to start treatment		2 (1.0–3.0)	3 (1.0–4.0)	0.11	-	-	-
Treatment with anti-CD agents, n (%)		28 (100.0)	43 (100.0)	0.40	-	-	-
	metronidazole	24 (85.7)	32 (74.4)		-	-	-
	vancomycin	3 (10.7)	10 (23.3)		-	-	-
	metronidazole + vancomycin	1 (3.6)	1 (2.3)		-	-	-
Charlson comorbidity index, median (IQR)		2 (0–3.75)	2 (1–3)	0.73	-	-	-
Inflammatory bowel diseases, n (%)		6 (21.4)	6 (14.0)	0.41	-	-	-
Chemotherapy for neoplasm, n (%)		3 (10.7)	4 (9.3)	1.00	-	-	-
White blood cell count, median (IQR)		9250 (7550–16,800)	9800 (6000–16,000)	0.76	-	-	-
Estimated glomerular filtration rate, median (IQR)		83.5 (46.6–111.7)	67.7 (41.2–106.2)	0.54	-	-	-
Serum albumin level, median (IQR)		2.4 (1.9–2.7)	2.4 (2.0–3.0)	0.52	-	-	-

NAAT, nucleic acid amplification test; CDI, *Clostridioides difficile* infection; and severe, severe to critical.

## Data Availability

Not applicable.

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
