# Peer review of "Impact of Nucleic Acid Amplification Test on Clinical Outcomes in Patients with Clostridioides difficile Infection"

_antibiotics, 2023, doi:10.3390/antibiotics12030428_

Round 1

Reviewer 1 Report

This is a well-written study.

Yamada et al. compared the clinical outcomes of CDI in a single-center in the era after implementing NAAT as an additional step to diagnose CDI to the outcomes in the era prior to the implementation. Yamada et al found a higher rate of clinical cure after the implementation of NAAT for CDI testing.

Few major notes:

1. It was very difficult to determine if EIA was used to detect GDH, the toxin A/B or both. Was GDH detected using EIA? If not, was this a 3 step diagnostic strategy (first GDH then Toxin by EIA then NAAT). On page 2, line 69, authors stated that EIA detected both GDH and TOXIN but figure 1. looks more like 2-3 different tests done first GDH then EIA followed by NAAT when indicated. I think the authors should make this point clearer for the readers.

2. I think the definition of severe CDI should be better characterized and referenced. MN severity criteria seems to be specific to Japan, international readers may not be familiar with these criteria.

3. May the authors clarify the sentence on page 4, lines 120-121 regarding 90-100% of patients with toxin + having high diagnostic rate. What was the gold standard test confirming that these GDH + toxin +, truly had CDI?

4. Can the authors clarify on what bases they determined that 63/84 in the toxin - in the pre-NAAT era and 23/49 patients with positive Toxin B by NAAT were non-CDI? I think this is very critical to explain as this paper's analysis depends on the number of patients in both groups who were eventually diagnosed with CDI.

5. It seems reasonable to still include the clinical cure as an outcome from table 2, even when it is already mentioned on page 4, lines 129-131.

6. Page 5, lines 160-162, is there a correlation between severity of CDI and lack of cure? If there is, can the authors reference? Many severe CDI can still be cured by a 14-day course of therapy with no diarrhea 2 days after finishing the treatment.

Thank you,

Reviewer 2 Report

The article “Impact of nucleic acid amplification test on clinical outcomes in patients with Clostridioides difficile infection” investigates the role of introduction of NAAT test on the clinical outcome of CDI over the period of 2 years in a hospital setup. The study shows that after introduction of NAAT based diagnosis of CDI clinical cure cases were significantly increased and hence can be used as an better and more sensitive approach as compared to EIA method. The article is well written However, the  following point should be address to improve the overall quality of article

Comments

1.     Line 50 please add reference

2.      What is the plausible reason for increase in clinically cure case after introduction of NAAT? Does NAAT resulted in early diagnosis and treatment which improved the overall outcome of the illness? Was the timeline for introduction of medicine/treatment monitored in both groups ?

Round 2

Reviewer 1 Report

I would like to thank the authors for the changes and amendments they added to the manuscript.

point 1: supplement table: can the authors check the values and the >, <, = signs, the signs should always precede the values. In the current table the values are sometimes before the signs and sometimes after the signs making the interpretation of these signs difficult.

Point 2: I think that somehow the authors should clarify somewhere in the methods section that patients who were transferred to other hospitals were excluded from the evaluation of the outcomes but still included when evaluating the diagnostic rates.

Thank you,

Reviewer 2 Report

The authors have successfully addressed all the points.

Round 3

Reviewer 1 Report

No further comments.

Thank you